# Aberrant Expression of RAD52, Its Prognostic Impact in Rectal Cancer and Association with Poor Survival of Patients

**DOI:** 10.3390/ijms21051768

**Published:** 2020-03-04

**Authors:** Vincent Ho, Liping Chung, Amandeep Singh, Vivienne Lea, Askar Abubakar, Stephanie H. Lim, Wei Chua, Weng Ng, Mark Lee, Tara L. Roberts, Paul de Souza, Cheok Soon Lee

**Affiliations:** 1School of Medicine, Western Sydney University, Campbelltown, NSW 2560, Australia; liping.chung@westernsydney.edu.au (L.C.); askar.abubakar@westernsydney.edu.au (A.A.); tara.roberts@westernsydney.edu.au (T.L.R.); p.desouza@westernsydney.edu.au (P.d.S.); soon.lee@westernsydney.edu.au (C.S.L.); 2Ingham Institute for Applied Medical Research, Liverpool, NSW 2170, Australia; Stephanie.Lim@health.nsw.gov.au (S.H.L.); wei.chua2@sswahs.nsw.gov.au (W.C.); 3Department of Anatomical Pathology, Liverpool Hospital, Liverpool, NSW 2170, Australia; amandeep.singh@health.nsw.gov.au (A.S.); vivienne.lea@health.nsw.gov.au (V.L.); 4Macarthur Cancer Therapy Centre, Campbelltown Hospital, NSW 2560, Australia; 5Discipline of Medical Oncology, School of Medicine, Western Sydney University, Liverpool, NSW 2170, Australia; 6Department of Medical Oncology, Liverpool Hospital, Liverpool, NSW 2170, Australia; weng.ng@health.nsw.gov.au; 7South Western Sydney Clinical School, University of New South Wales, Liverpool Hospital, Liverpool, NSW 2170, Australia; 8Department of Radiation Oncology, Liverpool Hospital, Liverpool, NSW 2170, Australia; mark.lee2@health.nsw.gov.au; 9Discipline of Pathology, School of Medicine, Western Sydney University, Campbelltown, NSW 2560, Australia

**Keywords:** DNA double-strand breaks (DSBs), RAD52, DNA damage response, rectal cancer, prognosis, biomarkers

## Abstract

The DNA damage response enables cells to survive and maintain genome integrity. RAD52 is a DNA-binding protein involved in the homologous recombination in DNA repair, and is important for the maintenance of tumour genome integrity. We investigated possible correlations between RAD52 expression and cancer survival and response to preoperative radiotherapy. RAD52 expression was examined in tumour samples from 179 patients who underwent surgery for rectal cancer, including a sub-cohort of 40 patients who were treated with neoadjuvant therapy. A high score for RAD52 expression in the tumour centre was significantly associated with worse disease-free survival (DFS; *p* = 0.045). In contrast, reduced RAD52 expression in tumour centre samples from patients treated with neoadjuvant therapy (*n* = 40) significantly correlated with poor DFS (*p* = 0.025) and overall survival (OS; *p* = 0.048). Our results suggested that RAD52 may have clinical value as a prognostic marker of tumour response to neoadjuvant radiation and both disease-free status and overall survival in patients with rectal cancer.

## 1. Introduction

Colorectal cancer (CRC) is the third most commonly diagnosed cancer, and is a major cause of cancer mortality worldwide. The global burden of CRC is expected to increase to more than 2.2 million new cases, with an estimated 1.1 million deaths by the end of 2030 [1]. Although surgical resection remains a primary treatment modality for all types of CRC, the surgical resectability of cancers of the colon and rectum determines their distinct clinical management. Limited access within the pelvic cavity and proximity to the mesorectal fascia and pelvic organs makes resection with clear margins of rectal cancers challenging, leading to a high risk of local recurrence [2]. 

Preoperative or neoadjuvant radiotherapy, in combination with 5-fluorouracil (5-FU), has produced significant improvements in clinical outcomes, such as decreased rate of local recurrence and reduced surgery-related morbidities, which has helped establish this approach as the standard of care for treatment of rectal cancer, especially for larger or slightly more advanced tumours, prior to resection [3]. However, there is considerable variation in individual response to preoperative chemoradiotherapy. Pathologic complete response, determined by the lack of a viable tumour after treatment, is seen in <30% of patients with rectal cancer, and approximately 40% of patients show no significant response to treatment [4,5,6]. 

Having the capacity to predict a pathological tumour response prior to treatment would help identify patients who would benefit the most from preoperative therapy. It would also help avoid the risk of toxicity in patients likely to be non-responsive to concurrent chemoradiotherapy. Molecular biomarkers are promising candidates for predicting treatment response at an early time point with sufficient sensitivity and specificity, since clinical biomarkers are not commonly in use. Although several predictive and prognostic biomarkers for CRC have been reported [7,8,9,10,11,12,13], none to date have been successfully implemented in a clinical setting, and the search for effective molecular predictors continues.

Although ionizing radiation (IR) in radiotherapy induces a variety of DNA lesions, including base damage and single strand breaks, double strand breaks (DSBs) are known to be the primary reason for tumour cell death [14,15]. Chemotherapy drugs such as 5-fluorouracil used in neoadjuvant therapy induce DNA damage and activate ATM/CHEK1 and/or CHEK2 in a dose-dependent manner. Chemotherapy/5-FU response can be variable and is affected by the mutation status of mismatch repair genes and TP53 as well as mutations to core DNA damage repair genes [16]. When DNA is damaged in cells, the DNA damage response (DDR) machinery is activated and induces checkpoints that stop cell cycle progression until the damaged DNA is repaired, thus ensuring genomic integrity. Repair of DSBs occurs primarily via three different pathways: homology-directed repair (HDR), non-homologous end joining (NHEJ), and the non-conservative annealing-based mechanism, single strand annealing (SSA). NHEJ, an error-prone mechanism, does not require sequence homology, involves minimal DNA processing for sealing the DSB, is active throughout the cell cycle, and is the primary repair pathway in vertebrate cells [17]. HDR is a relatively error-free mechanism that requires the use of homologous sequences to align DSB ends prior to ligation. This process is mediated by the generation of 3′ single-strand DNA (ssDNA) via a process called end resection, which involves the nucleolytic degradation of the 5′ strand of a DSB end. Strand invasion of the end resection intermediate into a homologous template then results in the synthesis of a nascent DNA strand that anneals to the other side of the DSB [18]. SSA is also a homology-based repair mechanism that involves the generation of 3′ ssDNA via end resection, which is then used for the annealing of homologous repeats flanking a DSB. SSA-mediated repair causes a deletion rearrangement between the repeats, thus resulting in the loss of genetic information [19].

The intrinsic radiosensitivity of tumour cells is influenced by their DNA repair capability [20]. If tumour cells effectively repair their damaged DNA, they develop a resistance to radiation, but a failure to do so will trigger programmed cell death [21]. Therefore, components of the DDR pathway have the potential to serve as key biomarkers or therapeutic targets for the treatment of rectal cancer.

RAD52 is a DNA-binding protein that is of prime importance for DSB repair via the SSA pathway. It binds the 3′ ssDNA end resection intermediate and mediates both a sequence homology search and DNA strand annealing in cis to form a DNA duplex, which is then processed by endonucleolytic cleavage, polymerase-mediated strand extension, and ligation to yield a repaired product [19,22]. Although yeast Rad52 is known to play an important role in Rad51-mediated HDR by facilitating the loading of Rad51 on replication protein A (RPA)-coated ssDNA, its role appears to be replaced by the breast cancer susceptibility protein 2 (BRCA2) in vertebrates [23,24]. However, synthetic lethality between RAD52 and BRCA2 deficiencies have indicated a functional redundancy between these two proteins [25]. Although RAD52 is known to play a key role in DNA repair, aiding maintenance of genomic integrity and cell viability, mutations associated with increased Rad52 expression have been linked to increased tumorigenesis and resistance to therapy [26,27].

It appears that RAD52 plays a key role in the maintenance of tumour genome integrity and furthermore is involved in the response to oncogene-induced DNA replication stress [26,28]. It is known that the single nucleotide polymorphisms (SNPs) in RAD52 can be linked to the risk of multiple cancers, including breast cancer, head and neck cancers, ovarian cancer, thyroid cancer and lung cancer [29,30,31,32,33]. High expressions of RAD52 have been found in lung squamous cancer and nasopharyngeal cancer tissue [34,35]. RAD52 is actively being investigated for its role in hepatocellular cancer pathogenesis [36]. 

Considering these varied roles of RAD52, we aimed to explore its potential as a prognostic marker for rectal cancer and as a predictor of tumour response to neoadjuvant therapy in rectal cancer patients.

## 2. Results

### 2.1. Study Population

A total of 179 patients (males: 119, 66.5%, and females: 60, 33.5%) were included in this study. Patient characteristics are listed in Table 1, based on an available clinical information. The median age was 71 years (range: 35–100 years). Of 177 patients, 61 (34.5%) were treated with radiotherapy, of which 40 (65.6%) received preoperative therapy (Appendix A). Patients were followed for a median period of 3.13 years (range: 0−12.6 years), and the median time to death was 2.9 years after surgery (range: 0−11.1 years).

### 2.2. Association between RAD52 Expression and Clinicopathological Features and Prognosis

We investigated the association between postoperative RAD52 expression levels and clinicopathological characteristics (Table 2). RAD52 expression levels in the tumour centre (TC) were significantly associated with lymph node (LN) involvement (*p* = 0.028). There were no significant differences in age, sex, tumour stage, grade, metastasis, vascular invasion, or perineural invasion between patients with low and high RAD52 protein expression. Interestingly, we found that RAD52 expression levels in the tumour periphery (TP) were associated with age group; patients older than 71 years exhibited significantly higher RAD52 expression than other age groups (*p* = 0.018). TC referred to the areas with highest mitotic activity at the centre of the rectal cancer, whilst TP referred to the most mitotically active areas at the outer invasive zone of the tumour. Figure 1 shows representative hematoxylin and eosin (H+E) and immunohistochemical staining of high and low RAD52 expression in rectal cancer tissues. A Kaplan–Meier survival analysis demonstrated that patients with high RAD52 protein expression in the TC had significantly worse disease-free survival (DFS; *p* = 0.045; Figure 2A) and overall survival (OS; *p* = 0.049; Figure 2B) than patients with low RAD52 expression. No significant differences in survival were seen between patients with high or low RAD52 protein expression in the TP (DFS, *p* = 0.445, Figure 2C; OS, *p* = 0.476; Figure 2D).

We next examined the status of the mismatch repair (MMR) pathway in patient samples by evaluating the association of MMR (MLH1, MSH2, MSH6 and PMS2) protein expression with RAD52 protein expression. An MMR gene defect leads from loss of the corresponding normal alleles in the tumours of carriers to loss of MMR function and results in an accumulation of mutations in microsatellites (MSI, microsatellite instability) in tumours. The loss of MMR protein expression can be detected by IHC and it can determine the identity of the mutated gene. In this study, all cases were positive for MLH1 and MSH2 expression; therefore, none of the cases were classified as microsatellite instability (MSI)-high (MMR-negative). The expression of MSH6 and PMS2 was absent in only 1.1% (2/174) and 4.7% (8/169) of cases, respectively, and expression of these proteins was not significantly associated with RAD52 expression in either TC or TP samples (Table 2). 

Using univariate Cox regression analysis, we found that high expression of RAD52 in the TC was significantly associated with reduced OS (Table 3). Additionally, multivariate Cox analysis demonstrated that RAD52 expression (*HR* = 1.525, 95% *CI* 0.788–2.950, *p* = 0.046), metastasis stage (*HR* = 3.215, 95% *CI* 1.265–8.169, *p* = 0.014), and vascular invasion (*HR* = 2.315, 95% *CI* 1.314–4.075, *p* = 0.004) remained significantly associated with OS (Table 3), implying that those markers together are strongly prognostic for OS in patients with metastatic rectal cancer. 

### 2.3. Correlation of RAD52 Expression with Survival Outcomes after Preoperative Neoadjuvant Therapy

Preoperative or neoadjuvant therapy is an accepted treatment for potentially improving survival in rectal cancer patients [9]. Treatments received by patients with rectal cancer in the whole cohort were listed in Appendix A. Of 40 patients who underwent preoperative treatment, twenty patients received the neoadjuvant chemo- and radiotherapy treatment and 17 patients received a combination of pre- and postoperative therapy (Appendix A). We firstly examined the possible association between clinicopathological characteristics and survival outcomes with preoperative RAD52 expression in a subset of 40 patients who received preoperative neoadjuvant radiotherapy. Of these 40 patients, 27 (67.5%) were male, 13 (33.5%) were female, and the median age was 67 years (range: 42−85 years) (Table 1). In the subgroup of patients who received neoadjuvant therapy, lower expression of RAD52 was significantly associated with worse DFS (Figure 3A, *p* = 0.025) and OS (Figure 3B, *p* = 0.048), implying that lower RAD52 expression results in increased resistance to radiotherapy. As an attempt, we also investigated the possible association between RAD52 and survival in rectal cancer patients treated with chemo- and radiotherapy. However, no significant association was found (Appendix A: DFS, *p* = 0.216 and OS, *p* = 0.178, respectively), probably due to a small sample set (*n* = 20).

### 2.4. Prognostic Implications of RAD52 Expression in Lymph Node (LN)-Positive Subgroup

In a sub-cohort analysis of patients who received preoperative neoadjuvant therapy, low RAD52 expression was associated with worse DFS (*p* = 0.012; Figure 3D) and OS (*p* = 0.014; Figure 3F) in patients with LN-positive tumours, but no association with survival outcome was observed in patients with LN-negative tumours (DFS, *p* = 0.317; OS, *p* = 0.244; Figure 3C,E). A multivariate Cox analysis in the LN-positive subgroup showed that expression of RAD52 correlated with OS (*HR* = 0.473, 95% *CI* 0.029-0.905, *p* = 0.021) (Table 3). This suggests that RAD52 expression has potential prognostic value in rectal cancer patients with LN-positive tumours. Further studies with larger patient numbers in lymph node involvement are needed to evaluate the value of neoadjuvant therapy (e.g., neoadjuvant short-course radiotherapy) in these particular groups.

## 3. Discussion

We have analysed the impact of RAD52 expression on survival outcomes of patients with rectal cancer. We also assessed correlations between RAD52 expression and tumour response to neoadjuvant radiotherapy, combined with 5-FU treatment. We found that high levels of post-operative RAD52 expression in the TC correlated with worse DFS and OS compared with low RAD52 expression. 

Activation and elevated expression of various DDR pathway proteins have been detected in several cancers, including CRC, at various stages of cancer development, including premalignant and metastatic stages [7,37,38,39]. DDR is activated by increased genomic instability, by activated stress signalling due to oxidative, replication, and mechanical stresses occurring during cancer growth, and by pro-oncogenic effects of certain DDR proteins [39,40]. Due to the fact that RAD52 is known to promote SSA-mediated DNA repair in mammalian cells [41], it is possible that high levels of RAD52 result in increased levels of SSA repair in rectal cancer cells. SSA repair is error-prone and results in deletions of intervening regions between homologous repeats; therefore, increased SSA could result in increased genomic instability. In fact, the high density of repetitive elements in mammalian genomes [42], and their enrichment in several cancer-related genes [43], have been known to facilitate increased cancer-associated rearrangements mediated by SSA [42,44]. Moreover, SSA between homologous repeats the flank two DSBs on different chromosomes and has been shown to cause translocations [45,46]. Increased genomic instability as a result of SSA may promote cancer growth and spread, which could explain our observation that high levels of RAD52 are associated with poor survival outcomes in rectal cancer patients. This effect may also underlie the correlation seen between high RAD52 levels and LN-positive tumours. 

High RAD52 levels in TC, but not in TP, correlated with poor prognosis. Differences in gene expression based on the tumour microenvironment have been previously reported for other cancers [47,48]. The TC is generally more prone to hypoxia and necrosis than the TP, which consists of rapidly dividing cells at the invasive edge. Therefore, differences in oxidative and replicative stress conditions between the TC and TP may be one factor underlying the TC-specific correlation, with RAD52 expression and patient survival outcomes in rectal cancer.

Although the MMR machinery in Saccharomyces cerevisiae influences SSA at multiple steps [49,50], MSH2, which is a key component of the MMR pathway, appears to be dispensable for SSA in mammalian cells [51]. Our results showed no significant association between RAD52 expression and MMR protein expression in rectal cancer tissues, which suggests that the prognostic implications of high RAD52 levels may operate independently of the MMR machinery in rectal cancer. However, further studies with larger patient numbers are certainly needed to estimate and characterize the association of RAD52 expression and MMR protein expression in tumours. 

IR in radiotherapy primarily induces DSBs, which, if effectively repaired by the DDR pathway, will cause tumour cells to become radiation resistant. Lack of efficient DNA repair can result in extreme genomic instability in cancer cells and ultimately trigger apoptosis and cell death, thus rendering them radiation sensitive. Further, as unrepaired DNA damage has been linked to the induction of type I interferon and consequent boosting of anti-tumour immunity, increased efficiency of repair can also limit post-radiation anti-cancer immune responses [52,53,54]. Therefore, the intrinsic radiosensitivity of tumour cells is strongly influenced by their DSB repair capability [20]. Several studies have shown that defects in the DDR pathway increase sensitivity to radiotherapy [55,56]. In fact, this has also been shown in CRC for DDR proteins such as RAD51, XRCC2, and MRE11/ATM [7,10,57]. Overexpression of RAD51 leads to poor survival outcomes in rectal cancer patients, which has been attributed to increased therapy resistance [7]. Interestingly, in our sub-cohort analysis of 40 patients that received neoadjuvant therapy, we found that lower levels of preoperative RAD52 expression were associated with worse DFS and OS, implying that lower RAD52 expression results in an increased resistance to radiotherapy. This is a similar result to our finding that low postoperative RAD50 expression was associated with worse DFS and OS in early tumour stage and low-grade rectal cancer [58]. However, we found that with RAD50, in combination with other proteins, the MRN complex (MRE11/RAD50/NBS1) behaves quite differently from RAD50 alone [59]. Rectal cancer with a high expression of MRN complex proteins was associated with worse DFS and OS. The implication here is that RAD50 has a distinct function in the DDR pathway beyond its defined role within the MRN complex [60]. RAD52 may similarly exert a unique function within the DDR pathway. 

Our results could be explained by the role that RAD52 plays in DSB repair. The two primary mechanisms contributing to IR-induced DSB repair include NHEJ and HDR [20]. RAD52 is a key component of SSA-mediated DNA repair; however, the role of SSA in the repair of randomly induced DSBs, such as those generated by IR, remains unknown. Synthetic lethality studies have shown that, in the absence of BRCA2, RAD52 is essential for DSB repair. BRCA2 is a critical component of the HDR pathway along with RAD51 [25], indicating that SSA functions as an alternative repair pathway for homologous recombination during DSB repair. It is well known that DDR is a complex network of cross-talking and redundant pathways, and dysfunction in any one DNA repair pathway may be compensated by the function of another, thus contributing to increased radiation resistance [61]. In the preoperative rectal cancer tissues with lower levels of RAD52 seen in our study, it is possible that HDR was the primary mechanism of IR-induced DSB repair, and HDR may have compensated for the decreased SSA activity in these cells. Due to the fact that HDR is a relatively error-free process, and repairs DSBs with high fidelity, it is likely that HDR is better than SSA in conferring resistance to tumour cells in response to IR. Interestingly, it has been shown in yeast that RAD51 actively inhibits SSA to prevent chromosomal translocations and, therefore, genomic instability [47]. Perhaps, activation of HDR at the expense of SSA downregulation may be a strategy employed by rectal cancer cells to achieve increased radiation resistance. In the case of bladder urothelial cancer [62], low mRNA levels of *RAD52* correlated significantly with a poor overall survival. It was speculated that low non-physiological levels of *RAD52* could promote a dysregulated HDR and this could possibly up-regulate error-prone backup repair pathways. Another study [63] has found that low RAD52 expression is associated with a poor response of cervical cancer cells to carboplatin. It was suggested that *RAD52* SNPs, either individually or collectively, could modify gene function and alter RAD52 protein expression levels, making the cervical cancer cell resistant to platinum agents.

Future work including sequencing studies should address the possibility that HDR is indeed dysregulated in rectal cancer tissues with low RAD52 expression.

## 4. Materials and Methods 

### 4.1. Patients

The study was approved by the South Western Sydney Local Health District Human Research Ethics Committee (HREC Reference: HREC/14/LPOOL/186; project number 14/103, approved on 25 May 2017). Overall, 179 out of 266 patients in the Anatomical Pathology Department had available tissues at the Liverpool Hospital, and were therefore chosen for this study. Specimens were collected from 179 patients who underwent surgery and radiotherapy for rectal cancer from 2000 to 2011. Patients were treated with either a 25 Gy dose, administered in five treatment fractions alone, or a 50.4 Gy dose administered in 28 fractions concurrently with 5-fluorouracil-based chemotherapy. Surgery consisted of a total mesorectal excision, as well as anterior or abdominoperineal resection. Surgeries were carried out during 2000–2011 by 23 surgeons. The follow-up included clinic visits, blood tests, colonoscopy, and imaging based on the recommendation of the treating specialist.

### 4.2. Response and Outcomes of Interest

Short-term response to radiotherapy was measured by tumour regression grade (TRG) according to the 7th edition of the American Joint Committee on Cancer Manual [64], which describes a scale of 0–3: 0 represents complete response without any viable malignant cells; 1 is a moderate response with small groups of malignant cells; 2 is a minimal response with fibrosis outgrowing residual malignancy; and 3 is a poor response with residual malignancy. Patients categorized with a TRG of 0, 1, or 2 were considered responders, while patients categorized with a TRG of 3 were considered non-responders. Variables included age, sex, pathological TNM stage, tumour grade, vascular invasion, perineural invasion, the level of tumour-infiltrating lymphocytes, and treatment. Outcomes were DFS, OS, and histologic TRG in the resected bowel. Long-term outcomes (DFS and OS) were assessed by Kaplan–Meier analysis. DFS was defined as the time from diagnosis to first recurrence, and OS was defined as the time from diagnosis to the last follow-up or death. 

### 4.3. Sample Preparation and Tissue Microarrays

Two cores (1 mm diameter) were obtained from each of five different sampling sites from pre- and post-operative rectal cancer formalin-fixed tumour samples. These sites were: tumour centre (TC); tumor periphery at invasive edge (TP); normal mucosa close/adjacent to the tumour; normal mucosa well away from the tumour; and the involved lymph nodes. The corresponding H+E sections were reviewed to localize the most representative areas of tumour and normal colorectal mucosa in tissue samples. These samples were then transferred into pre-drilled wells in tissue microarray blocks using the Beecher Manual Tissue Microarrayer (Beecher Instruments Inc., Sun Prairie, WI, USA). The blocks were then mounted on slides for immunohistochemical analysis. 

### 4.4. Immunohistochemistry

Immunohistochemical staining preparation was performed as described previously [59]. Prepared slides were incubated with a monoclonal anti-RAD52 primary antibody, clone 5H9 (1:500 dilutions, Life Span Bioscience #LS-B1709; Seattle, WA, USA) for 60 min at room temperature. After washing with Tris-buffered saline with Tween-20, slides were incubated for 15 min with DAKO mouse linker, rinsed, and incubated for 30 min with an anti-mouse secondary antibody. A mixture of EnvisionTM FLEX DAB+Chromogen DM827 and EnvisionTM FLEX Substrate Buffer DM823 (DAKO) was used as a substrate for development. Finally, slides were counterstained with hematoxylin, washed with cold water, and then dipped 10 times in Scott’s Bluing solution. Slides were rinsed immediately with cold water before dehydration and mounting.

Samples were independently scored by at least two pathologists. RAD52 expression was scored as the product of the staining percentage and intensity, as previously described [59]. Intensity was graded as: 0, negative; 1, weak; 2, moderate; or 3, strong. The percentage of positive cells was graded as: 0 (<5%), 1 (5–25%), 2 (26–50%), 3 (51–75%), or 4 (>75%). These two measures were multiplied to obtain weighted scores ranging from 0–12. All tumour samples were categorized into either a low expression group (score range: 0–5) or a high expression group (score range: 6–12), determined by the median of the score range. Assessment of mismatch repair proteins was based on positive or negative staining for MLH1, MSH2, MSH6, and PMS2, irrespective of the proportion of cells stained.

### 4.5. Statistical Analysis

Statistical analyses were performed with SPSS for Windows 25.0 (IBM Corporation, Armonk, NY, USA). A survival analysis was conducted both for the entire cohort and, separately, in patients who received preoperative radiotherapy. In addition, further subgroup analysis was conducted for early tumour stage and low-grade tumours as covariates. Univariate and multivariate analyses were performed using Kaplan–Meier curves and Cox’s proportional hazards survival modelling for RAD52 protein expression from the cancer core and periphery. Covariates were sex, age, TNM stage, tumour grade, lymph node involvement, metastasis stage, vascular invasion, perineural invasion, prior chemotherapy, and prior radiotherapy. Univariate analysis was performed using the Mann–Whitney U test. *P* < 0.05 was considered statistically significant.

## 5. Conclusions

Our findings show that RAD52 expression is a potential prognostic factor for rectal cancer, with elevated levels of RAD52 predicting poor survival outcome. RAD52 may also hold potential as a predictive factor for rectal tumour response to preoperative radiotherapy, including response of LN-positive tumours, with low levels of RAD52 correlating with poor survival outcome. 

Future studies should address the mechanisms underlying the prognostic implications of RAD52 in rectal cancer. Such studies would provide greater understanding of the complex interplay between the different types of DSB repair pathways, which would likely contribute to the design of new treatment strategies that can be combined with radiotherapy to improve survival outcomes in various types of cancer.

## Figures and Tables

**Figure 1 ijms-21-01768-f001:**
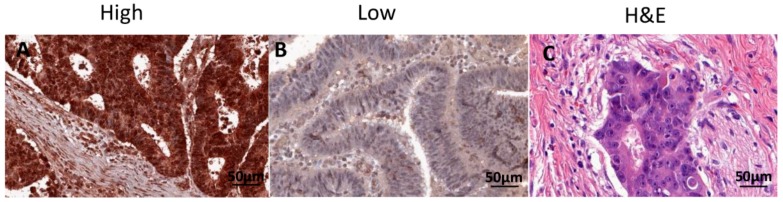
Representative immunohistochemical staining and hematoxylin and eosin (H+E) of RAD52 expression in rectal cancer samples. Staining for each protein was scored as high or low as described in the Methods. Representative examples of typical nuclear staining of RAD52 scored as high (**A**) and low (**B**) expression in tumour cells and corresponding H+E staining (**C**) are shown (40× magnification, scale bar: 50 µm).

**Figure 2 ijms-21-01768-f002:**
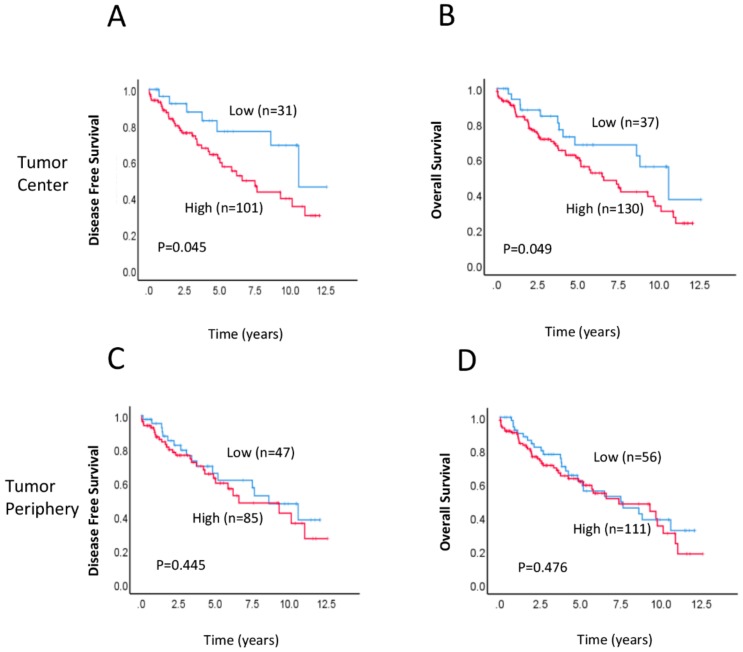
Association between survival and RAD52 expression in the tumour centre (TC) and tumour periphery (TP). (**A**–**D**) Kaplan–Meier survival analysis illustrating disease-free survival (DFS) (A,C) and overall survival (OS) (B,D) of patients with high (red line) and low (blue line) RAD52 protein expression in the TC (A,B) and TP (C,D).

**Figure 3 ijms-21-01768-f003:**
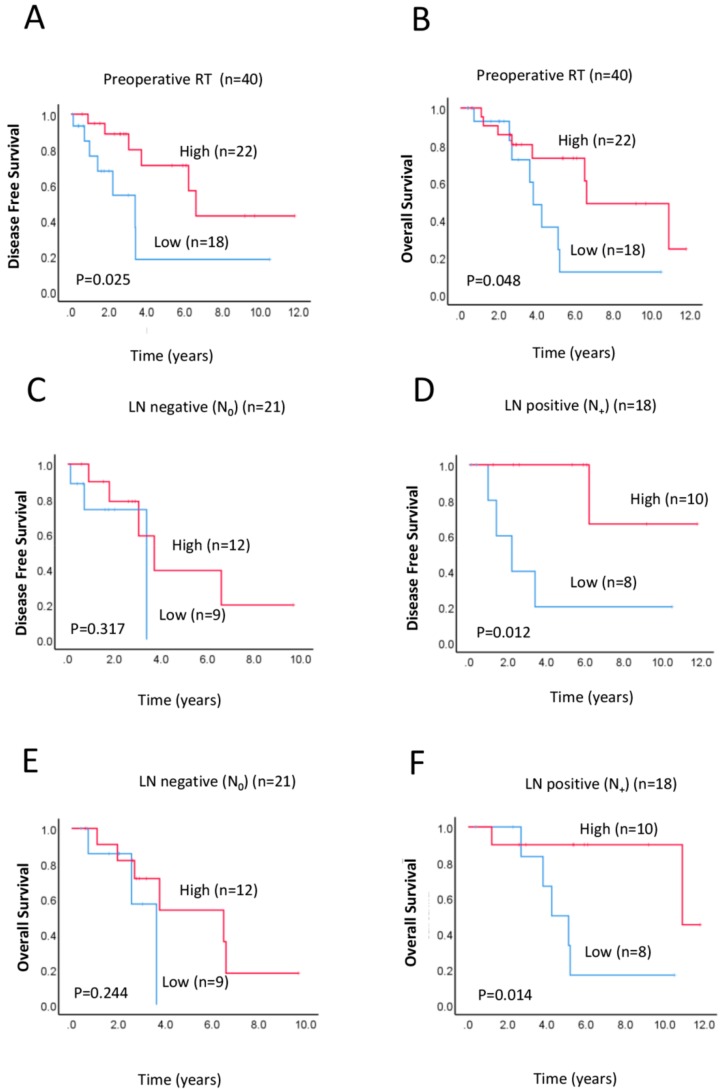
Relationship between preoperative RAD52 expression in the TC and survival, and RAD52 expression according to preoperative radiotherapy and lymph node (LN) involvement. (**A**,**B**) Kaplan–Meier survival analysis of DFS (A) and OS (B) in preoperative radiotherapy patient groups with low (blue line) or high (red line) RAD52 expression. (**C**–**F**) Kaplan–Meier survival analysis of DFS (C,D) and OS (E,F) in patients with high (red line) or low (blue line) RAD52 expression, in LN-negative (C,E) and LN-positive (D,F) rectal cancers.

**Table 1 ijms-21-01768-t001:** Patient characteristics.

Variables	All Patients (%)	Preoperative Neoadjuvant-Therapy Group
Total, *n*	179	40
Age median	71	67
Gender		
Male	119 (66.5)	27 (67.5)
Female	60 (33.5)	13 (32.5)
Tumour stage		
T1, T2	58/177 (32.8)	13/40 (32.5)
T3, T4	121/177 (67.2)	27/40 (67.5)
Node stage		
N0	92/173 (53.2)	23/40 (57.5)
N1, N2	81/173 (46.8)	17/40 (42.5)
Metastasis stage		
M0	152/161(94.4)	38/39 (97.4)
M1	9/161 (5.9)	1/39 (2.6)
Grade		
1, 2	167/179 (93.3)	38/40 (92.7)
3	12/179 (6.7)	2/40 (7.3)
Vascular invasion		
Absent	137/177 (77.4)	35/40 (87.5)
Present	40/177 (22.6)	5/40 (12.5)
Perineural invasion		
Absent	155/177 (87.6)	33/40 (82.5)
Present	22/177 (12.4)	7/40 (17.5)
Chemoradiotherapy		
Total	61/177 (34.5)	-
Neoadjuvant	40/61 (65.6)	-
Adjuvant	21/61 (34.4)	0/40 (0)
Recurrence		
Absent	93/148 (62.8)	22/38 (57.9)
Present	55/148 (37.2)	16/38 (42.1)
Tumour regression grade		
0, 2 (good response)	N/A	6/37 (16.2)
3 (poor response)	N/A	31/37 (83.8)

**Table 2 ijms-21-01768-t002:** Associations between clinicohistopathological data and RAD52 expression in the tumour centre and tumour periphery.

Variables	Subgroups	Tumour Centre			Tumour Periphery		
		Low (%)	High (%)	*p* value	Low (%)	High (%)	*p* value
Sex	Male	71.4	64.4	0.709	64.4	67.2	0.405
	Female	28.6	35.6		35.6	32.8	
Age	≤71	31.0	51.9	0.489	44.1	49.6	0.018
	>71	69.0	48.1		55.9	50.4	
Tumour stage	T1, T2	45.0	29.3	0.065	41.4	28.4	0.134
	T3, T4	55.0	70.7		58.6	71.6	
Node stage	Negative	58.5	49.6	0.028	63.8	46.1	0.320
	Positive	41.5	50.4		36.2	53.9	
Metastasis stage	M0	94.4	93.7	0.453	96.3	93.4	0.984
	M1	5.6	6.3		3.7	6.6	
Grade	1, 2	88.1	94.1	0.673	91.5	93.3	0.196
	3	11.9	5.9		8.5	6.7	
Vascular invasion	Absent	87.8	73.1	0.184	82.8	73.7	0.053
	Present	12.2	26.9		17.2	26.3	
Perineural invasion	Absent	92.7	85.1	0.842	86.2	87.3	0.208
	Present	7.3	14.9		13.8	12.7	
Adjuvant therapy	No	75.0	68.1	0.456	52.4	71.4	0.458
	Yes	25.0	31.9		47.6	28.6	
Neoadjuvant therapy	No	70.3	77.5	0.165	64.9	82.7	0.365
	Yes	29.7	22.5		35.1	17.3	
MSH6	Negative	0	2.2	0.198	3.6	0.8	0.354
	Positive	100	97.8		96.4	98.2	
PMS2	Negative	7.9	4.7	0.088	7.3	4.4	0.445
	Positive	92.1	95.3		92.7	95.6	

Bolded values: *p* <0.05; MSH6: MutS protein homolog; PMS2: PMS1 homolog 2.

**Table 3 ijms-21-01768-t003:** Cox regression analyses of RAD52 expression in TC with overall survival.

Variables	Univariate				Multivariate *		
	*n* (%)	*HR*	95% *CI*	*p* Value	*HR*	95%	*p* Value
RAD52							
High	76.5	1.711	0.915–3.200	0.040	1.525	0.788–2.950	0.046
Low	23.5						
Sex							
Male	66.1	0.912	0.549–1.574	0.721			
Female	33.9						
Age							
≤71	46.9	1.192	0.726–1.951	0.484			
>71	53.1						
Tumour stage							
T1–2	32.9	1.962	1.118–3.411	0.091			
T3–4	67.1						
Node stage							
Negative	51.7	1.075	0.663–1.741	0.077			
Positive	48.3						
Metastasis stage							
M0	94.4	4.652	1.922–11.260	0.001	3.215	1.265–8.169	0.014
M1	5.6						
Grade							
1, 2	92.7	1.003	0.403–2.496	0.995			
3	7.3						
Vascular invasion							
Absent	76.6	2.659	1.577–4.482	0.001	2.315	1.314–4.075	0.004
Present	23.4						
Perineural invasion							
Absent	86.9	1.794	0.933–3.448	0.08			
Present	13.1						
Adjuvant therapy							
No	69.7	0.482	0.222–1.047	0.065			
Yes	30.3						
Neoadjuvant therapy							
No	75.9	0.437	0.546–1.682	0.588			
Yes	24.1						
LN-negative ^†^	53.8				0.869	0.025–0.503	0.939
LN-positive ^†^	46.2				0.473	0.029–0.905	0.021

Bolded values: *p* <0.05; *HR*; hazard ratio; *CI*: confidence interval; ^†^ denotes interaction; LN: lymph node; * Multivariate cox proportional-hazards model was established using three covariates including RAD52, metastasis status and vascular invasion as predictive factors.

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
