# Peer review of "Aberrant Expression of RAD52, Its Prognostic Impact in Rectal Cancer and Association with Poor Survival of Patients"

_ijms, 2020, doi:10.3390/ijms21051768_

Round 1
Reviewer 1 Report
The authors report a protein that correlates to poor survival in patients. RAD52 has been previously reported in the last decade to associate with survival in other cancer types, such as patients with cervical cancer. Here, the authors sought to highlight the clinical significance of RAD52 in rectal cancer, and conclude that RAD52 may have clinical value as a prognostic marker of tumor response to neoadjuvant radiation and both disease-free and overall survival in patients with rectal cancer. Some comments:
- Table 1: please provide scale bars on figure 1
- Prior literature of RAD52 correlations on the prognosis of other cancer types seem to be severely lacking from the introduction.
- The authors state that all tumor samples were categorized into either a low expression group (score range: 0–5) or a high expression group (score range: 6–12). Please elaborate on how the score range is determined.
- How do the findings of RAD52 in rectal cancer compare with prior literature on cervical and bladder cancer?These could be highlighted in the discussion
- The authors state that high RAD52 levels in TC but not in TP correlated with poor prognosis. Please discuss the possible explanations and impact of this finding.
- Several acronyms were not explained in the text, e.g., TC and TP
Author Response
Note: Page numbers refer to the revised manuscript with all changes accepted.
Reviewer 1.
General: The authors report a protein that correlates to poor survival in patients. RAD52 has been previously reported in the last decade to associate with survival in other cancer types, such as patients with cervical cancer. Here, the authors sought to highlight the clinical significance of RAD52 in rectal cancer, and conclude that RAD52 may have clinical value as a prognostic marker of tumor response to neoadjuvant radiation and both disease-free and overall survival in patients with rectal cancer. Some comments:
- Table 1: please provide scale bars on figure 1
Response: Clarified. All scale bars are included in the revised manuscript (see page 4, line 139).
- Prior literature of RAD52 correlations on the prognosis of other cancer types seem to be severely lacking from the introduction.
Response: We thank the reviewer for pointing out this. Now we have included more literature on RAD52 correlations on the prognosis of other cancer types and this is embedded into the Introduction (see below).
It appears that RAD52 plays a key role in the maintenance of tumour genome integrity and furthermore is involved in the response to oncogene-induced DNA replication stress [26,28]. It is known that single nucleotide polymorphisms (SNPs) in RAD52 can be linked to the risk of multiple cancers, including breast cancer, head and neck cancers, ovarian cancer, thyroid cancer and lung cancer [29, 30,31,32,33]. High expressions of RAD52 have been found in lung squamous cancer and nasopharyngeal cancer tissue [34,35]. RAD52 is actively being investigated for its role in hepatocellular cancer pathogenesis [36].
- The authors state that all tumor samples were categorized into either a low expression group (score range: 0–5) or a high expression group (score range: 6–12). Please elaborate on how the score range is determined.
Response: Clarified (see page 12, line 145).
- How do the findings of RAD52 in rectal cancer compare with prior literature on cervical and bladder cancer?These could be highlighted in the discussion.
Response: We thank the reviewer for pointing this out. We have included more literature on the relevance of RAD52 to both cervical and bladder cancers and also in relation to low RAD52 expression and this is embedded in the Discussion (see below)
In the case of bladder urothelial cancer [59] low mRNA levels of RAD52 correlated significantly with a poor overall survival. It was speculated that low non-physiological levels of RAD52 could promote a dysregulated HDR and this could possibly up-regulate error-prone backup repair pathways. Another study [60] has found that low RAD52 expression is associated with a poor response of cervical cancer cells to carboplatin. It was suggested that RAD52 SNPs, either individually or collectively, could modify gene function and alter RAD52 protein expression levels, making the cervical cancer cell resistant to platinum agents.
5.The authors state that high RAD52 levels in TC but not in TP correlated with poor prognosis. Please discuss the possible explanations and impact of this finding.
Response: Clarified, we have included more discussion in relation to RAD52 expression levels in TC correlated with poor prognosis as TC referred to the areas with highest mitotic activity particularly in this revised manuscript (see page 10 from line 43).
- Several acronyms were not explained in the text, e.g., TC and TP
Response: Clarified (see page 4, lines 119 and 123)
Reviewer 2 Report
In this study, the authors investigated the association between RAD52 expression in tumor samples from 179 rectal cancers and response to preoperative radiotherapy and patients' survival. The authors report that high levels of RAD52 expression in the tumor center correlate with poor disease-free and overall survival. In contrast, for the neoadjuvant subgroup of patients, reduced expression of RAD52 is associated with a worse prognosis. No significant association was found between RAD52 expression and survival in cancer patients treated with chemo- and radiotherapy. Though the sample size was small and caution is warranted in interpreting these results, experiments and data analysis were in general well carried out. While further studies with larger sample sizes are needed to develop a predictive marker for the response, this work provides preliminary evidence for the possible predictive value of RAD52 expression in rectal cancer.
Minor issues:
Page 4, line 117: the second part of the sentence ", suggesting that differences in oxidative and replicative stress conditions between the TC and TP may be one factor underlying the TC-specific correlation with RAD52 expression" is vague and can be moved to and explained in the discussion section. The number (N) of cases in each subgroup should be presented in tables. Information in table S1 and Fig. S1 should be also moved to the main textAuthor Response
Reviewer 2.
General: In this study, the authors investigated the association between RAD52 expression in tumor samples from 179 rectal cancers and response to preoperative radiotherapy and patients' survival. The authors report that high levels of RAD52 expression in the tumor center correlate with poor disease-free and overall survival. In contrast, for the neoadjuvant subgroup of patients, reduced expression of RAD52 is associated with a worse prognosis. No significant association was found between RAD52 expression and survival in cancer patients treated with chemo- and radiotherapy. Though the sample size was small and caution is warranted in interpreting these results, experiments and data analysis were in general well carried out. While further studies with larger sample sizes are needed to develop a predictive marker for the response, this work provides preliminary evidence for the possible predictive value of RAD52 expression in rectal cancer.
Minor issues:
Page 4, line 117: the second part of the sentence ", suggesting that differences in oxidative and replicative stress conditions between the TC and TP may be one factor underlying the TC-specific correlation with RAD52 expression" is vague and can be moved to and explained in the discussion section. The number (N) of cases in each subgroup should be presented in tables. Information in table S1 and Fig. S1 should be also moved to the main text
Response: We appreciate the reviewer’s interest in our work. As suggested, we have edited the section for discussing in relation to the TC-specific correlation with RAD52 expression (see page 10 from line 43). The numbers of subjects in each group were also presented in this revised version (Figure 2 and Figure S1, respectively). More information for the supplementary Table S1 and Figure S1 has been presented in the revised manuscript.